# Relation-Constrained Decoding for Text Generation

**Xiang Chen**,* **Zhixian Yang**,* **Xiaojun Wan**
Wangxuan Institute of Computer Technology, Peking University
Center for Data Science, Peking University
The MOE Key Laboratory of Computational Linguistics, Peking University
caspar@pku.edu.cn, yangzhixian@stu.pku.edu.cn, wanxiaojun@pku.edu.cn

## Abstract

The dominant paradigm for neural text generation nowadays is seq2seq learning with large-scale pretrained language models. However, it is usually difficult to manually constrain the generation process of these models. Prior studies have introduced *Lexically Constrained Decoding (LCD)* to ensure the presence of pre-specified words or phrases in the output. However, simply applying lexical constraints has no guarantee of the grammatical or semantic relations between words. Thus, more elaborate constraints are needed. To this end, we first propose a new constrained decoding scenario named *Relation-Constrained Decoding (RCD)*, which requires the model's output to contain several given word pairs with respect to the given relations between them. For this scenario, we present a novel plug-and-play decoding algorithm named **RE**lation-guided probability **S**urgery and b**E**am **AL**location (RESEAL), which can handle different categories of relations, e.g., syntactical relations or factual relations. Moreover, RESEAL can adaptively "reseal" the relations to form a high-quality sentence, which can be applied to the inference stage of any autoregressive text generation model. To evaluate our method, we first construct an RCD benchmark based on dependency relations from treebanks with annotated dependencies. Experimental results demonstrate that our approach can achieve better preservation of the input dependency relations compared to previous methods. To further illustrate the effectiveness of RESEAL, we apply our method to three downstream tasks: sentence summarization, fact-based text editing, and data-to-text generation. We observe an improvement in generation quality. The source code is available at https://github.com/CasparSwift/RESEAL.

## 1 Introduction

Incorporating complex manual constraints into neural text generation is a challenging research topic. One of the most important manual constraints is the *relation constraint*, i.e., to guarantee that two pre-specified words must appear in the generated text and keep the given relation between them. Such relation constraints have various applications. For instance, data-to-text generation [11, 20] and fact-based text editing [17] aim to ensure the presence of given facts (entities and relations between them) in the output. Moreover, in sentence summarization task [36], there are some key semantic relations that must be preserved to ensure the fluency and factual constituency of the summaries.

The most prominent paradigm for text generation is seq2seq learning by finetuning the large-scale pretrained models [21, 34] and obtaining the outputs by beam search in an autoregressive manner. However, this paradigm often fails to satisfy the complex constraints because there is no explicit mechanism to enforce these constraints. To tackle this problem, previous works [15, 16, 31] propose *Lexically Constrained Decoding (LCD)* to preserve some given keywords in the output. However,

---

*Equal contribution.

36th Conference on Neural Information Processing Systems (NeurIPS 2022).

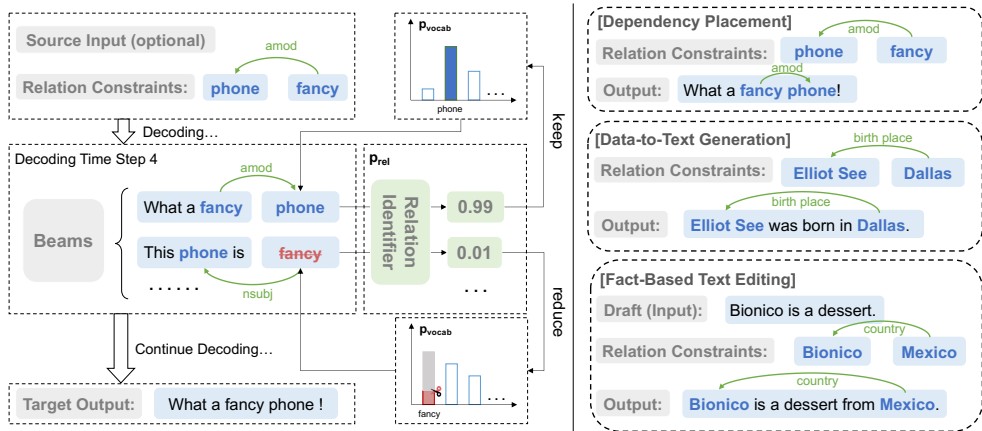

Figure 1: The framework of our proposed RESEAL. Given the relation constraints *(phone, amod, fancy)*, RESEAL operates the next-token probability $p_{\text{vocab}}$ according to the probabilities $p_{\text{rel}}$ produced by a relation identifier. RESEAL will relatively maintain or reduce the $p_{\text{vocab}}$ of those candidate words that meet the relation constraints in an adaptive way. In this case, the probability of "phone" is maintained since it can form an "amod" relation with "fancy". On the contrary, in another beam, the probability of "fancy" is cut down since it will form a wrong relation "nsubj" with "phone". Note that "amod" denotes adjectival modifier and "nsubj" denotes nominal subject.

simply utilizing these lexical constraints still struggles to ensure the word relation constraints. Therefore, another form of constrained decoding is needed to handle the relation constraints.

In this paper, we propose a new constrained decoding scenario named *Relation-Constrained Decoding (RCD)*. Specifically, we adopt the triplets *(head, relation, tail)* as the *relation constraints*. At the decoding stage, we aim to force the model output to include these relation constraints. The right part of Figure 1 shows three instances for RCD with different relation types. In an evident way, satisfying relation constraints requires satisfying corresponding lexical constraints. A straightforward solution to this problem is to generate a set of candidate sentences using any LCD algorithm to ensure the keyword preservation, and then rerank them by the number of relation constraints they have met. However, this approach requires to first generate a number of whole sentences, which is inefficient and inflexible. To this end, we propose RESEAL (**RE**lation-guided probability **S**urgery and b**E**am **AL**location), a relation-guided decoding algorithm for RCD that can dynamically adjust the choice of words during decoding. RESEAL modifies a conventional LCD method, i.e., Dynamic Beam Allocation (DBA) algorithm [31], and incorporates a high-quality external *relation identifier* to identify the presence of relation constraints. As illustrated in the left part of Figure 1, based on the relation identifier, RESEAL dynamically adjusts the probability of the candidate constrained words in the generation process[2].

Among all categories of word relations, the dependency relation is most basic, standard and representative which has various public available datasets for evaluation. Therefore, in this paper, we mainly focus on the dependency relation scenario of RCD. To illustrate the effectiveness of our proposed RESEAL, we construct a benchmark from publicly available treebanks [39] that contain sentences and their dependency trees annotated by human. We randomly sample a subset of dependency triplets as the input constraints and regard the original sentences as reference outputs. We call this task "Dependency Placement". Experiment results show that our method outperforms baselines and LCD methods on dependency coverage (the ratio of satisfied relation constraints for dependency). After that, to showcase the applicability of this work, we further explore some potential applications of RCD. We conduct extensive experiments on three downstream tasks: sentence summarization (with dependency relations), fact-based text editing (with relations between two entities in the knowledge graph), and data-to-text generation (with relations extracted from knowledge bases). Across different tasks, we observe a consistent improvement over the strong baselines.

---

[2]Note that the performance of relation identifier is crucial to the generation quality. It's not so difficult to train an accurate relation identifier. We will further discuss this external dependence issue in Appendix D.3.

To sum up, the contributions of our work are three-fold: (i) We propose Relation-Constrained Decoding (RCD), a scenario for constrained text generation, and construct its benchmarks. To the best of our knowledge, we are the first to study this problem. (ii) We design RESEAL, a decoding algorithm that can generate high-quality sentences that meet relation constraints. (iii) The experimental results on the RCD task and downstream tasks including sentence summarization, fact-based text editing and data-to-text generation show the effectiveness of RESEAL.

## 2  Problem Formulation

In this section, we first formulate the Relation-Constrained Decoding (RCD) problem. For the text generation tasks, given an input sequence $X = (x_1, x_2, ..., x_N)$, where $N$ is the input sequence length, $x_i \in \mathcal{V}_\mathcal{S}$ and $\mathcal{V}_\mathcal{S}$ is the source vocabulary, the model generates a sequence $Y = (y_1, y_2, ..., y_M)$, where $M$ is the output sequence length, $y_i \in \mathcal{V}_\mathcal{T}$ and $\mathcal{V}_\mathcal{T}$ is the target vocabulary. The conditional probability of $Y$ given $X$ and model parameter $\theta$ can be calculated as follows:

$$p(Y|X;\theta) = \prod_{t=1}^{M} p(y_t|y_{<t}, X; \theta). \tag{1}$$

Eq. 1 usually acts as an objective for beam search. In this paper, we denote each relation constraint as a triplet $(h, r, \tau)$, where $h$ is the *head*, $\tau$ is the *tail*, and $r$ is the relation between them. Given an unordered relation constraints set $C = \{(h_l, r_l, \tau_l)\}_{l=1}^{L}$, where $L$ is the number of constraints and $h_l, \tau_l \in \mathcal{V}_\mathcal{T}$, we aim to make the output $Y$ satisfy the constraints in $C$ as much as possible. For the model's output $Y$, we denote $C'(Y) = \{(h_l', r_l', \tau_l')\}_{l=1}^{M}$ be the relation triplets of $Y$, and then we propose to jointly optimize Eq. 1 and $|C \cap C'(Y)|$ as a novel objective for RCD.

## 3  Methodology

Algorithm 1 gives an overview of RESEAL. To start the decoding, the decoder input is initialized with a single $\langle\texttt{BOS}\rangle$ token. At each time step $t$, the model maintains the $k$-best candidate sentences, where $k$ is the beam size. The decoder produces the distribution $p_{\text{vocab}}(w|y_{<t}, X; \theta)$ for each token $w$ in the target vocabulary $\mathcal{V}_\mathcal{T}$ (line 3). Each candidate has different $p_{\text{vocab}}$ respectively to produce $k|\mathcal{V}_\mathcal{T}|$ candidates, then we can select top-$k$ candidates from them by the cumulative log probability (line 5). The decoding ends when candidates contain $k$ finished sentences (line 6-7). Different from the standard beam search, RESEAL follows a two-step approach as follows:

---
**Algorithm 1** RESEAL (overview)

---
**Input:** Max sequence length $N$, beam size $k$, relation constraints $C$, relation identifier $R$, enc_inputs.
**Output:** Output sequence.
1: Initialize $k$ candidates and dec_inputs
2: **for** time step $t$ **in** $[1, N]$ **do**
3:      $p_{\text{vocab}} \leftarrow$ MODEL(enc_inputs, dec_inputs)
4:      $\tilde{p} \leftarrow$ PROB_SURGERY($p_{\text{vocab}}$, candidates, $C$, $R$)
5:      candidates $\leftarrow$ RG_TOPK($\tilde{p}$, candidates, $C$)
6:      **if** have finished $k$ sentences **then**
7:          **break**
8: **return** candidate with highest score

---

**Step 1: Probability Surgery (line 4)**    RESEAL operates the produced probability distributions according to the result of a *relation identifier*, which serves as an explicit signal to guarantee the presence of relation constraints.

**Step 2: Relation-Guided Top-K (line 5)**    To satisfy lexical constraints, we replace the standard Top-$K$ operation by the one used in DBA [31]. Furthermore, to satisfy relation constraints, RESEAL dynamically allocates beam by the results of *relation identifier* instead of the number of satisfied lexical constraints used in DBA.

If removing line 4 and replacing line 5 with a normal Top-$K$, RESEAL is equivalent to the standard beam search. We will describe the aforementioned two steps in detail in the rest of this section.

### 3.1  Probability Surgery

Algorithm 2 (line 1-6) shows the process of probability surgery. At time step $t$ during decoding, the model predicts the next token probability $p_{\text{vocab}}(w|y_{<t}, X; \theta)$. To provide an external signal to guarantee the presence of relation constraints, we calculate another probability distribution $p_{\text{rel}}(w|y_{<t}, C)$

---

**Algorithm 2** Probability Surgery and RG-Top-K

---

1: **function** PROB_SURGERY($p_{\text{vocab}}$, candidates, $C$, $R$)
2:     **for all** candidate **in** candidates **do**
3:         **for all** unmet lexical constraints $w$ of candidate **do**
4:             sent ← candidate.sentence (i.e., $y_{<t}$) + $w$
5:             Get $p_{\text{trans}}, p_{\text{type}}$ by $R$ and sent, and then calculate $p_{\text{rel}}(w|y_{<t}, C)$ by Eq. 3.
6:         Calculate and normalize $\tilde{p}$ by $\widetilde{p}(w|y_{<t}, X, C; \theta) \propto g(p_{\text{rel}}(w|y_{<t}, C)) \cdot p_{\text{vocab}}(w|y_{<t}, X; \theta)$.
        **return** $\tilde{p}$ of every candidate
7: **function** RG_TOPK($\tilde{p}$, candidates, $C$)
8:     candidates ← Generate_candidates_by_DBA($\tilde{p}$, candidates)
9:     Initialize relation_counts, bank, and pruned_candidates.
10:     **for all** candidate **in** candidates **do**
11:         Get $n_i$, which is the number of correct relation constraints in this candidate.
12:         Update relation_counts, and add the candidate to bank $n_i$.
13:     bank_sizes ← Beam_Allocate_by_DBA(relation_counts)
14:     **for** $j$ **in** $[0, |C|]$ **do**
15:         Add top-$K$ candidates in bank $j$ to pruned_candidates, where $K =$ bank_sizes$[j]$.
        **return** pruned_candidates

---

by the external relation identifier. The $p_{\text{rel}}(w|y_{<t}, C)$ indicates the probability that the token $w$ satisfies the relation constraints $C$ given previous decoding result $y_{<t}$. The core idea of our proposed probability surgery is to combine $p_{\text{vocab}}$ and $p_{\text{rel}}$ together to produce an augmented distribution $\tilde{p}$, and then use $\tilde{p}$ instead of $p_{\text{vocab}}$ to choose the words.

We first give a formal definition of $p_{\text{rel}}(w|y_{<t}, C)$. The $p_{\text{rel}}$ depends on two factors: (1) the transition probability $p_{\text{trans}}(y_j, y_i)$, the probability that $y_j$ is the head of $y_i$, (2) the relation type probability $p_{\text{type}}(y_j, r, y_i)$, the probability that the relation between $y_i$ and $y_j$ falls in type $r$. Both $p_{\text{trans}}$ and $p_{\text{type}}$ can be obtained during decoding by a *relation identifier*. For dependency relations, the relation identifier can be a left-to-right dependency parser [9] to better fit the left-to-right manner of autoregressive decoding. For relations between entities, the relation identifier can be any relation extraction model. The function of relation identifier is to predict the head of $y_i$, which produces a series of $p_{\text{trans}}(y_j, y_i)$. Additionally, the relation identifier also predict the relation types, which produces a series of $p_{\text{type}}(y_j, r, y_i)$. Note that if $y_j$ is not the head of $y_i$, $p_{\text{type}}(y_j, r, y_i) = 0$ for all types $r$.

Given the incomplete output sequence $y_{<t} = (y_1, y_2, ..., y_{t-1})$ at time step $t$ and the next token $w$, we choose the relation constraints related to $w$ in $C$. Let $C_{\text{head}}(w, t), C_{\text{tail}}(w, t)$ denote the subset of relation constraints $C$ at time step $t$ where the $w$ serves as the head or the tail, respectively:

$$
\begin{aligned}
C_{\text{head}}(w, t) &= \{(w, r, y)|(w, r, y) \in C \land y \in y_{<t}\}, \ \forall w \in \mathcal{V}_{\mathcal{T}}. \\
C_{\text{tail}}(w, t) &= \{(y, r, w)|(y, r, w) \in C \land y \in y_{<t}\}, \ \forall w \in \mathcal{V}_{\mathcal{T}}.
\end{aligned}
\tag{2}
$$

We can calculate $p_{\text{rel}}(w|y_{<t}, C)$ as follows[3]:

$$
p_{\text{rel}}(w|y_{<t}, C) = \frac{1}{Z_{w,t}} \left\{ \sum_{\substack{(w,r,y) \in \\ C_{\text{head}}(w,t)}} [p_{\text{trans}}(w, y) + p_{\text{type}}(w, r, y)] + \sum_{\substack{(y,r,w) \in \\ C_{\text{tail}}(w,t)}} [p_{\text{trans}}(y, w) + p_{\text{type}}(y, r, w)] \right\},
\tag{3}
$$

where $Z_{w,t} = 2[|C_{\text{head}}(w, t)| + |C_{\text{tail}}(w, t)|]$ is a normalizing factor. Consequently, the augmented distribution can be calculated as follows:

$$
\widetilde{p}(w|y_{<t}, X, C; \theta) \propto g(p_{\text{rel}}(w|y_{<t}, C)) \cdot p_{\text{vocab}}(w|y_{<t}, X; \theta),
\tag{4}
$$

where $g : [0, 1] \to (0, 1]$ is a gate function to transform $p_{\text{rel}}$ to a weight of $p_{\text{vocab}}$. We aim to increase the weight when the $p_{\text{rel}}$ increases, so $g$ cannot be a monotonically decreasing function. Moreover, the output of this function should not be exactly zero, because assigning zero probability will make the log-likelihood of the whole sentence be negative infinite. Based on these requirements, inspired

---

[3]Note that $p_{\text{rel}}(w|y_{<t}, C) = 1$ if $C_{\text{head}}(w, t) = C_{\text{tail}}(w, t) = \emptyset$. Apart from that, we use additive form of $p_{\text{rel}}$ instead of a multiplicative manner, we further discuss this in Appendix A.

by Schick et al. [37], we adopt this parameterized form of $g$ in this paper:

$$g(p_{\text{rel}}) = \begin{cases} e^{-\lambda(1-p_{\text{rel}})} & \text{if } p_{\text{rel}} < \rho \\ 1 & \text{otherwise} \end{cases} \tag{5}$$

where $\lambda, \rho$ are the hyperparameters and $\lambda > 0, \rho \in (0,1]$. $\lambda$ controls the decay of output value. $\rho$ is a threshold for the probability. We fix $\rho = 0.5$ in this paper. If the $p_{\text{rel}}$ of a word $w$ is greater than or equal to this pre-specified threshold $\rho$, it is confident enough to consider $w$ satisfies the relation constraints. Thus we set $g(p_{\text{rel}}) = 1$ for this situation to preserve the $p_{\text{vocab}}$ of $w$. On the contrary, when $p_{\text{rel}}$ is close to 0, $g(p_{\text{rel}})$ will be close to a relatively small value $e^{-\lambda}$, which indicates that the model will be less likely to choose the words violating relation constraints.

## 3.2 Relation-Guided Top-K

To ensure the presence of lexical constraints, we adopt the Top-$K$ operation in DBA [31]. DBA firstly generates a set of candidates and then selects $k$ of them through beam allocation. The candidate set generated by DBA (line 8) is the union of three sets: (1) the normal Top-$K$ tokens, (2) all unsatisfied lexical constraints, and (3) the single-best token for each hypothesis in the beam. After that, DBA groups together the candidates with the same number of satisfied lexical constraints into some *banks* and selects a different number of candidates from different banks. The candidates with fewer lexical constraints will have more chances to be selected. However, the original DBA is not aware of the relations between words. Since we have already processed the candidate sentences by relation identifier in probability surgery, we can now use the processed result to guide the bank allocation. As illustrated in Algorithm 2, we propose to use the number of correct relation constraints of $i$-th candidates (line 10-12) to divide the banks, rather than the number of satisfied lexical constraints used by DBA. This modification can jointly consider both lexical and relation constraints, because one relation constraint is equivalent to two lexical constraints and their relation.

# 4 Experiments on Dependency Placement

There are many kinds of word relations in natural language, so it is necessary to showcase the performance of our proposed RESEAL on different relations. Since syntactic dependency structures serve as the principle of how words are combined to form sentences, dependency relations can be the most basic and important for text generation. Thus, in this section, we mainly focus on the dependency relation scenario of RCD, and evaluate RESEAL on the **Dependency Placement** task. Besides, we will conduct extensive experiments on three downstream tasks in Section 5.

## 4.1 Task and Dataset

We first define **Dependency Placement** task: given the constraints $C$ of dependency relations, output a fluent sentence $Y$ which appropriately places these constraints. The model input $X$ is optional, which can be a single $\langle \text{BOS} \rangle$ token, or a sequential transformation of $C$ to provide necessary information. We then construct the dataset for dependency placement task from the English-EWT [39] corpus[4], which contains 16,621 sentences with dependency annotations and standard train/dev/test set split. For each sentence with $m$ words, we randomly sample $n$ dependency triplets $\{(h_i, r_i, \tau_i)\}_{i=1}^{n}$ as the given constrains $C$, where $n < m$. The original sentences serve as references. We refer to this dataset as English-EWT-Dep. More details about English-EWT-Dep can be found in Appendix B.

## 4.2 Evaluation Metrics

In this section, we discuss the appropriate metrics for the dependency placement task to evaluate an RCD algorithm (including but not limited to our proposed RESEAL). Simply using the automatic evaluation to compare the system outputs with the references is not suitable, because there are too many sentences that can be the correct answer given several dependencies. We mainly focus on whether the relation constraints are satisfied when examining an RCD algorithm. To this end, the output sentences should be processed again by an accurate external parser[5]. This parser should

---

[4] https://universaldependencies.org/
[5] Another choice is to process the outputs by human, which is too expensive.

Table 1: Evaluation result for dependency placement task. "Reference" denotes evaluating the ground truth sentences, which can be viewed as an approximated upper bound of this task. Despite that the BLEU-4 and METEOR is not so accurate to evaluate this task, we still provide it for reference only.

| Method | Stanza | | spaCy | | BLEU-4↑ | METEOR↑ | PPL↓ | Word%↑ |
|---|---|---|---|---|---|---|---|---|
| | UC↑ | LC↑ | UC↑ | LC↑ | | | | |
| Base [21] | 80.52 | 69.69 | 81.04 | 71.02 | 11.92 | 20.12 | 865.40 | 97.11 |
| Rerank ($k = 20$) | 84.32 | 74.86 | 84.04 | 74.93 | 11.66 | 20.18 | 346.44 | **99.88** |
| CGMH [26] | 39.46 | 25.70 | 37.50 | 24.69 | 1.47 | 14.50 | 2341.83 | 96.20 |
| X-MCMC [13] | 51.78 | 37.36 | 52.30 | 37.99 | 4.62 | 17.04 | 513.18 | 99.86 |
| X-MCMC-C [13] | 58.17 | 44.90 | 58.90 | 46.23 | 6.39 | 17.65 | 557.58 | **99.88** |
| DBA [31] | 79.54 | 67.39 | 79.78 | 68.47 | 11.47 | 20.12 | 318.67 | 99.82 |
| DDBA [25] | 79.22 | 68.72 | 79.96 | 70.11 | 12.22 | 20.12 | 796.76 | 97.01 |
| NeuroLogic [24] | 82.47 | 71.72 | 83.03 | 72.87 | 12.23 | 20.13 | 436.27 | 98.93 |
| RESEAL | **86.45** | **79.26** | **86.73** | **80.66** | **12.62** | **20.40** | **260.80** | 99.60 |
| Reference | 86.80 | 81.49 | 90.50 | 86.49 | 100.00 | 100.00 | 527.35 | 100.00 |

preferably be different from the one used in an RCD algorithm, which can better examine the generalization ability across the parsers. In this paper, we use two widely-used dependency parsers provided by `Stanza` [32] and `spaCy`[6] for evaluation. Let $C^{(i)}$ denote the relation constraints of $i$-th output $Y^{(i)}$ in test set. Let $C'(Y^{(i)})$ denote the dependency relation triplets obtained by the external parser. Let $W^{(i)}$ and $W'(Y^{(i)})$ denote the sets if we omit the dependency relation $r$ of $C^{(i)}$ and $C'(Y^{(i)})$. Similar to the unlabeled/labeled attachment score (UAS/LAS) used in dependency parsing, we can define the unlabeled/labeled coverage (UC/LC) as follows:

$$\text{UC} = \frac{\sum_i |W^{(i)} \cap W'(Y^{(i)})|}{\sum_i |W^{(i)}|}, \quad \text{LC} = \frac{\sum_i |C^{(i)} \cap C'(Y^{(i)})|}{\sum_i |C^{(i)}|}. \tag{6}$$

Moreover, we report the BLEU-4 [30], METEOR [2], GPT-2 [33] perplexity (PPL) and word coverage (the proportion of lexical constraints that are satisfied).

### 4.3 Baselines

Since there are no existing work about dependency placement, we design some straightforward baselines to compare with our method:

- **Base**: Use BART$_{\text{large}}$ [21] as the backbone, and then funetine it on English-EWT-Dep. The input is the concatenation of the triplets of $C$ separated by special token #. For example, if $C = \{(h_1, r_1, \tau_1), (h_2, r_2, \tau_2)\}$, the input sequence $X = h_1 \# r_1 \# \tau_1, h_2 \# r_2 \# \tau_2$. The target output is the reference. During decoding, we use standard beam search with beam size $k = 20$.

- **CGMH** [26]: Use MCMC sampling to generate a sentence by modifying it. We use BERT$_{\text{large}}$ [6] to produce its replacement probability, and GPT-2$_{\text{large}}$ [33] as its language model.

- **X-MCMC** [13]: Improve CGMH by using XLNet [46]. **X-MCMC-C** adds a classifier to instruct the X-MCMC models where and how to modify the candidate sentences.

- **DBA** [31]: Use DBA algorithm to decode on the **Base** model with beam size $k = 20$.

- **DDBA** [25]: A denoised variant of DBA by filtering noisy constraints.

- **NeuroLogic** [24]: A LCD algorithm which support complex lexical constraints in Conjunctive Normal Form (CNF).

- **Rerank**: Preserve all $k$ sentences generated by DBA, and select the sentence that satisfies the most relation constraints using left-to-right parser [9].

We report the results when applying RESEAL to the **Base**. More details can be found in Appendix C.1.

---

[6]https://explosion.ai/blog/ud-benchmarks-v3-2#project

Table 2: Result of ablation study for dependency placement task. We remove a single component from the full algorithm to study the individual effect. "w/o prob" denotes without probability surgery (but still with RG-Top-K), "w/o RG-Top-$K$" denotes using the way of original DBA to allocate beam (but still with probability surgery). "word+rel" denotes using the number of satisfied lexical and (dependency) relation constraints to allocate beam.

| Method | Stanza | | spaCy | | BLEU-4↑ | METEOR↑ | PPL↓ | Word%↑ |
| | UC↑ | LC↑ | UC↑ | LC↑ | | | | |
|---|---|---|---|---|---|---|---|---|
| RESEAL | **86.45** | **79.26** | **86.73** | **80.66** | **12.62** | **20.40** | 260.80 | 99.60 |
| w/o prob | 81.70 | 70.91 | 82.12 | 72.21 | 11.83 | 20.19 | **256.88** | **99.80** |
| w/o RG-Top-$K$ | 82.12 | 74.27 | 83.03 | 75.49 | 11.89 | 20.21 | 361.96 | 99.76 |
| word+rel | 82.40 | 74.51 | 83.34 | 75.42 | 11.08 | 19.81 | 452.50 | 99.72 |

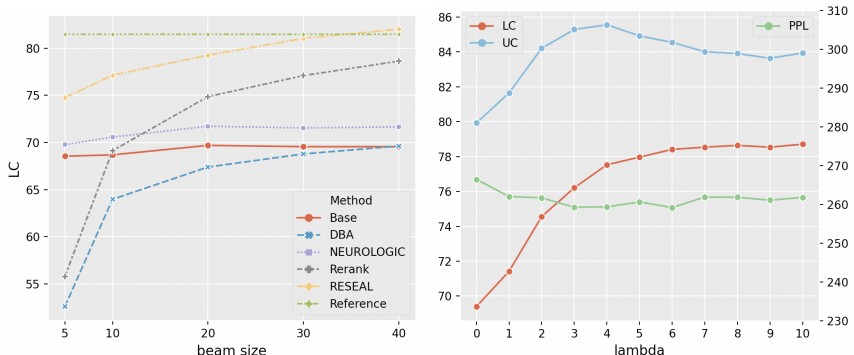

Figure 2: The result when altering the value of $\lambda$ and beam size $k$.

## 4.4 Discussion

**Results**  Table 1 shows the results for dependency placement. We find that the sampling-based method [13, 26] can achieve better word coverage, but the low UC/LC scores demonstrate that they fail to enforce the relation constraints. RESEAL achieves best UC/LC among all the methods with significant decline of the PPL and competitive word coverage. Specifically, RESEAL gains an improvement of 1.15 on BLEU-4 and 11.87%/12.19% (Stanza/spaCy) on LC compared to DBA. These results illustrate the weakness of existing LCD algorithms to correctly place the multiple relation constraints. Some LCD methods (DBA, DDBA, NeuroLogic) would forcibly add the unsatisfied lexical constraints into the candidate word set. This is the key to ensuring the presence of lexical constraints. However, doing this fails to consider the relations between words, thus would make the generated sentence less fluent. Apart from that, RESEAL outperforms Rerank on almost all the metrics. An intrinsic reason for this may be that RESEAL can dynamically adjust the word selection according to the parsing result during decoding, but Rerank can not make a choice until finishing all the sentences.

**Ablation Study**  Table 2 shows the results of ablation study. We observe a decrease of 0.79 on BLEU-4 and 8.35%/8.45% (Stanza/spaCy) on LC by removing probability surgery. We also observe a decrease of 0.73 on BLEU-4 and 4.99%/5.17% (Stanza/spaCy) on LC by removing RG-Top-$K$. Probability surgery enables the tokens with correct dependencies to enter the candidate set. RG-Top-$K$ dynamically allocates the beam according to the parsing result to satisfy more relation constraints. These two components are both crucial. Furthermore, we find that adopting another beam allocation strategy ("word+rel" in Table 2) also hurt the performance.

**Impact of Hyperparameters**  For the dependency placement task, the performance would benefit from a larger beam size, which is consistent with the observations of Post and Vilar [31]. The left of Figure 2 shows the labeled coverage (LC) as a function of beam size by different methods on the test set. We observe that RESEAL can achieve better LC scores with smaller beam sizes when compared with Rerank. Apart from that, the decay factor $\lambda$ introduced in Section 3.1 is another important hyperparameter of RESEAL. We investigate its influence on the model performance. Figure 2 shows

Table 3: Evaluation result for sentence summarization experiments.

| Methods | Gigaword | | | DUC2004 | | | MSR-ATC | | |
|---|---|---|---|---|---|---|---|---|---|
| | RG-1 | RG-2 | RG-L | RG-1 | RG-2 | RG-L | RG-1 | RG-2 | RG-L |
| SEASS [49] | 46.86 | 24.58 | 43.53 | 29.21 | 9.56 | 25.51 | 25.75 | 10.63 | 22.90 |
| Keyword [22] | 47.14 | 25.06 | 44.39 | - | - | - | - | - | - |
| SemSum [18] | - | - | - | 31.00 | 11.11 | 26.94 | 33.82 | 17.08 | 30.62 |
| BART [21] | 50.14 | 27.37 | 46.69 | 31.38 | 11.43 | 27.51 | 40.39 | 22.09 | 35.32 |
| BART+RESEAL | **50.73** | **27.84** | **47.18** | **32.67** | **11.63** | **28.38** | **43.77** | **25.28** | **37.78** |
| BART+RESEAL (gold) | 53.74 | 31.00 | 48.99 | 35.53 | 12.41 | 28.68 | 69.42 | 41.76 | 51.71 |

Table 4: Examples with RESEAL and other baselines. The dependency relation constraints are (*thought*, `ccomp`, *'s*) and (*'s*, `nsubj`, *charges*).

| Method | Generated Sentences |
|---|---|
| Base | I **thought** the **charges** would be $ 5,000, but they were $ 10,000. |
| DBA | I **thought** the **charges** would be $ 10,000, but it**'s** $ 20,000. |
| Rerank | I **thought** they were going to charge me, but there**'s** no **charges**. |
| RESEAL | I **thought** there**'s** some **charges**, but I was wrong. |
| Ref. | I always **thought** there**'s** no custom **charges** for gifts. |

the results when varying $\lambda$ from 0 to 10 based on the validation set. If we set $\lambda$ as a relatively large value ($> 4$), the performance of RESEAL tends to be stable. Specifically, $\lambda = 0$ is equivalent to removing probability surgery (from Eq. 5), which will result in worse performance.

**Case Study**    Table 4 shows the sentences generated by all listed methods for dependency placement task. The **Base** model may omit some lexical constraints. DBA and Rerank satisfy all the lexical constraints, but they cannot correctly handle the relations between them. Rerank may generate sentences with repetitions (e.g., the word "charge"). RESEAL can produce sentences that are close to the grammatical structure of the references.

**More Discussions**    In Appendix D, we discuss the limitations of RESEAL, including the impact of dependency parsers, time complexity and external dependence issue. In Appendix E, we provide more cases for dependency placement task. In Appendix F, we discuss the social impact of this work.

## 5  Experiments on Downstream Tasks

To further explore the effectiveness of our proposed RESEAL, we conduct experiments on the three downstream tasks: sentence summarization, fact-based text editing, and data-to-text generation. Intuitively, RESEAL can aid these tasks. For the sentence summarization, RESEAL may help to preserve the important relations in the source sentence. For the rest two tasks, RESEAL can help to incorporate the given factual relations.

### 5.1  Sentence Summarization

**Dataset**    We conduct experiments on English Gigaword dataset [36], which contains about 3.8M training sentence pairs. We use the validation and test set provided by Zhou et al. [49] with 8,000 and 2,000 sentence pairs, respectively. Following previous work, we also evaluate our model on the test set of DUC2004 [29] (with 500 input sentences) and MSR-ATC [42] (with 785 input sentences).

**Dependency Prediction**    To apply RESEAL to sentence summarization, we first need to obtain reasonable dependency triples to construct relation constraints. In this paper, we train a vanilla BERT-base-uncased [6] model to predict which dependencies should be present in the target output. Firstly, we parse the sentences in the dataset by the left-to-right parser [9]. Then we use the intersection of dependencies of source and target sentences as the ground truth to train the dependency predictor. For each triplet $(h, r, \tau)$, we concatenate the contextual embedding of $h$, $\tau$ and label embedding of $r$ to perform binary classification. More details can be found in Appendix C.2.

Table 5: Evaluation result of WebEdit dataset.

| Methods | BLEU-4 | SARI | KEEP | ADD | DELETE |
|---|---|---|---|---|---|
| EncDecEditor [17] | 71.03 | 69.59 | 89.49 | 43.82 | 75.48 |
| FactEditor [17] | 75.68 | 72.20 | 91.84 | 47.69 | **77.07** |
| Seq2Seq | 82.96 | 73.74 | 93.62 | 64.56 | 63.05 |
| Seq2Seq+RESEAL | **84.12** | **78.33** | **96.07** | **69.63** | 69.29 |

**Results**    Following previous work [18, 49], we report ROUGE F1 [23] on Gigaword and MSR-ATC, and ROUGE recall on DUC2004. Table 3 show the results. BART consistently outperforms the previous models without pretraining across different datasets. Over the strong BART baseline, RESEAL can achieve better ROUGE scores on all datasets with predicted relation constraints. We also investigate the upper bound of BART+RESEAL by using the gold relation constraints, which shows that there is room for improvement with more accurate dependency predictors.

Table 6: Evaluation result for WebNLG test set.

| Methods | BLEU-4 |
|---|---|
| Castro Ferreira et al. [4] | 51.68 |
| Moryossef et al. [27] | 47.24 |
| Zhao et al. [48] | 52.78 |
| Harkous et al. [12] | 52.90 |
| Nan et al. [28] | 45.89 |
| T5-small [34] | 56.34 |
| T5-small + RESEAL | **56.87** |
| T5-base [34] | 59.17 |
| T5-base + RESEAL | **59.59** |

## 5.2 Fact-Based Text Editing

**Dataset**    Fact-based Text Editing is a novel task proposed by Iso et al. [17]. Given some triplets (facts) from knowledge graphs and a draft text, this task aims to revise the draft text to contain these facts. We adopt the WebEdit dataset provided by Iso et al. [17], which contains 181K/23K/29K instances as train/valid/test set. Note that this dataset can be viewed as a natural scenario for RCD because both the relation constraints (facts) and model input (draft text) are provided. More importantly, based on the results of error analysis, the models trained on this dataset still suffer from missing facts and incorrect relations (see "Qualitative evaluation" section in [17]). RESEAL may alleviate these problems by explicitly enforcing facts and relations.

## 5.3 Data-to-Text Generation

**Models and Results**    For the relation identifier, we train a simple BiLSTM [14] encoder with biaffine attention [8] on the training set of WebEdit (See Appendix C.3.1 for more details). We use EncDecEditor and FactEditor reported by Iso et al. [17] as our baseline models. EncDecEditor is an encoder-decoder model based on LSTM, with two separate encoders for facts and drafts and a decoder for generating revised texts. FactEditor shares the same encoders with EncDecEditor but has a novel decoder that doesn't follow the conventional autoregressive decoding manner. Thus we can only apply RESEAL to the EncDecEditor. For a fair comparison, we do not use pretrained models and keep the same architecture setting as EncDecEditor. However, the source code and the training details of EncDecEditor are unreleased, thus we reimplement EncDecEditor using our own training configuration (See Appendix C.3.2). We denote this model as Seq2Seq. Table 5 shows the experiment results of WebEdit. Following Iso et al. [17], we report the BLEU-4 and SARI [45] score (the average F1-score for keep, add and delete operations). Owing to the difference of training settings, Seq2seq can achieve significant improvement on BLEU-4 (+7.28) and SARI (+1.54) compared to FactEditor. Based on this result, we further adopt RESEAL on Seq2Seq and then observe a substantial improvement on BLEU-4 (+1.16) and SARI (+4.59). The above results demonstrate that RESEAL can achieve better facts and relations preservation over the strong Seq2Seq baseline.

**Dataset**    Data-to-text generation is another direct application for RESEAL. In this paper, we adopt WebNLG dataset [11] which provides facts as inputs and sentences containing these facts as outputs. We use the data provided by Ribeiro et al. [35] which contains 18,102/872/1,862 instances as train/valid/test set. Each test instance has 1-3 references.

**Models and Results**    For WebNLG dataset, the setting of the relation identifier is the same as that for WebEdit. Table 6 shows the experiment results of WebNLG dataset. We adopt our RESEAL on

T5 [34] and report the BLEU-4 score for evaluation. We observe an improvement of 0.53 BLEU-4 on T5-small and 0.42 BLEU-4 on T5-base. The detailed experimental settings can be found in Appendix C.4.

## 6   Related Work

**Lexically Constrained Decoding (LCD)**     Prior explorations for LCD can be summarized into four categories. The first line of studies has proposed some model-agnostic methods which only modify the decoding process. They are independent from the training. Hokamp and Liu [15] propose the grid beam search (GBS) algorithm, a modification to beam search to impose the lexical constraints. Post and Vilar [31] propose the dynamic beam allocation (DBA) algorithm with less time complexity. Vectorized DBA [16] and Denoised DBA [25] are two different DBA variants. The second line of studies requires some modifications to the training process. Augmenting the training data with the lexical constraints is a general approach [5, 7, 40]. Another branch of previous works focuses on adding additional structure to the model [41, 43, 44]. The fourth line of studies applies Markov Chain Monte Carlo (MCMC) to constrained text generation in a refinement manner [13, 26, 38]. Different from these methods, we do not only focus on the isolated lexical constraints. We propose to adopt relation constraints to consider the relationship between words.

**Dependency-Guided Generation**     Dependency is a natural way to represent the syntactic or semantic relations between words, so it can be used to guide the text generation. There are few works exploring this. Filippova and Strube [10] propose to compress the dependency graph to guide the sentence fusion. Akoury et al. [1] propose to predict a chunked syntactic parse tree and then generate tokens conditioned on the parse. Jin et al. [18] encode the dependency relations by a graph encoder to improve sentence summarization. Casas et al. [3] propose a language model where the generation is driven by the expansion over the dependency parse tree. Yang and Wan [47] propose a dependency modeling objective to incorporate dependency knowledge. However, one drawback of these methods is the limitation of interpretability and controllability since they only use the dependency as a latent variable during training, and cannot explicitly control the generation at the inference stage.

## 7   Conclusion

In this paper, we explore the Relation-Constrained Decoding (RCD), a new decoding scenario with a more complex definition of constraints. We propose RESEAL, a novel algorithm for RCD, which can be applied to the decoder of different models to preserve specific relation constraints. We examine two different experiment settings of RCD: dependency placement and downstream tasks. For dependency placement, we construct the benchmark for dependency placement, and the experiment results show the strength of RESEAL for satisfying relation constraints. Furthermore, we apply RESEAL to three downstream tasks as extended experiments for practical applications. Extensive experiments demonstrate the effectiveness and universality of our method.

## Acknowledgment

This work was supported by National Key R&D Program of China (2021YFF0901502), National Science Foundation of China (No. 62161160339), State Key Laboratory of Media Convergence Production Technology and Systems and Key Laboratory of Science, Technology and Standard in Press Industry (Key Laboratory of Intelligent Press Media Technology). We appreciate the anonymous reviewers for their helpful comments. Xiaojun Wan is the corresponding author.

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
