## A    More Explanations about RESEAL

Both $C_{\text{head}}(w,t)$ and $C_{\text{tail}}(w,t)$ can be empty set as $C$ can be arbitrary. Noted that in this part we simply sum over all $p_{\text{trans}}$ and $p_{\text{type}}$ to produce $p_{\text{rel}}$, which is different from the method we use when combining $p_{\text{vocab}}$ and $p_{\text{rel}}$. The reason why we use this form is that if substituting Eq. 5 and 3 into Eq. 4, we can derive a decomposed form:

$$g(p_{\text{rel}}(w|y_{<t},C)) = \left\{ \prod_{\substack{(w,y,r)\in \\ C_{\text{head}}(w,t)}} [g(p_{\text{trans}}(w,y)) \cdot g(p_{\text{type}}(w,y,r))] \cdot \right. \\ \left. \prod_{\substack{(y,w,r)\in \\ C_{\text{tail}}(w,t)}} [g(p_{\text{trans}}(y,w)) \cdot g(p_{\text{type}}(y,w,r))] \right\}^{Z_{w,t}} . \tag{7}$$

Eq. 7 indicates that we assign high probability to $w$ at time step $t$ if and only if all the relation constraints in $C_{\text{head}}(w,t) \cup C_{\text{tail}}(w,t)$ are satisfied. Any violation of the constraints will result in a small value of $g(p_{\text{rel}}(w|y_{<t},C))$ and reduce the augmented probability $\tilde{p}$ of $w$ according to Ed. 4.

## B    Dataset Construction for English-EWT-Dep

In this section, we give more details about the data construction process of English-EWT-Dep introduced in Section 4.1. We only consider the dependency relations whose modifier is an *adjective, noun, or verb* and randomly sample parts of them (at a ratio of $40\%$). The sentences which are shorter than 4 words are omitted. Some sentences containing personal information (e.g. phone number or address) are also discarded. The statistics of the English-EWT-Dep dataset is as follows:

Table 7: Statistics of the English-EWT-Dep dataset.

|  | train | dev | test |
|---|---|---|---|
| Sentence | 10119 | 1454 | 1407 |
| Avg word count | 19.39 | 15.98 | 16.19 |
| Avg number of dependencies | 2.43 | 2.04 | 2.04 |
| Max number of dependencies | 18 | 13 | 11 |

## C    Implementation Details

We provided the implementation details for the four tasks included in this work: dependency placement, sentence summarization, fact-based text editing and data-to-text generation. Note that in this paper, we conduct all the experiments on a NVIDIA A40 GPU with 40 GB memory.

### C.1    Dependency Placement

For the **Base** model of dependency placement task, we use BART$_{\text{large}}$ [21] as our pretrained model. We use label smoothing of $0.1$ and optimize the model by Adam [19]. The learning rate is 2e-5 with a warmup rate of $0.1$. The weight decay is $0.1$. We train the **Base** model for 10 epochs with a batch size of $32$.

### C.2    Sentence Summarization

For the sentence summarization task, we also use BART$_{\text{large}}$ [21] as our pretrained model. The hyperparameter setting is similar to the dependency placement except that we only train for $5$ epochs with a batch size of $100$. The ROUGE evaluation option for GigaWord and MSR-ATC is -m -n 2 -w 1.2. The ROUGE evaluation option for DUC2004 is -m -b 75 -n 2 -w 1.2.

For the dependency predictor used in this task, we adopt a BERT-base-uncased [6] model. Since the dependency is a word-level relation and BERT uses sub-word tokenization, we use the mean vector of the start and the end token as the representation $f(w)$ of a word $w$:

$$f(w) = \frac{1}{2}[\text{BERT}(\text{start token of } w) + \text{BERT}(\text{end token of } w)]. \tag{8}$$

We use an additional embedding layer with the dimension of 200 to represent the dependency label $r$. Let $f_L(r)$ denotes the representation of $r$, the feature of $(h, r, w)$ is defined as the concatenation of $f(h)$, $f(w)$ and $f_L(r)$.

We optimize the dependency prediction model by Adam [19]. The learning rate is 2e-5 with a warmup rate of 0.1. The weight decay is 0.1. We train this model for 10 epochs with a batch size of 200. After training, for each sentence in test set, we select the dependency relation constraints with predicted probability larger than a threshold $\rho = 0.6$.

It's worth noting that our reported performance of sentence summarization can be further improved by training a more accurate dependency predictor, or using a single keyword extractor and preserving the corresponding dependencies of predicted keywords. We leave more in-depth explorations as future work.

### C.3    Fact-Based Text Editing

Different from sentence summarization task, fact-based text editing does not require to predict the relation constraints, because the relation triplets have been provided in the dataset. Therefore, to adopt RESEAL, we only need to train the relation identifier and the LSTM [14] seq2seq model.

### C.3.1    Relation Identifier for Fact-Based Text Editing

The structure of relation identifier for fact-based text editing is borrowed from the biaffine dependency parser [8]. We use a single layer LSTM with single direction. We use two separate MLPs with one hidden layer to extract the features of heads and tails, respectively. Finally, we use a biaffine attention module to classify the relation types. Moreover, following the setting of Iso et al. [17], we use a simple embedding lookup table to embedding the input words. The detailed hyperparameter setting is shown as follows:

| | |
|---|---|
| Embedding size | 300 |
| LSTM hidden size | 300 |
| MLP hidden size | 512 |
| MLP output size | 300 |
| Number of relation categories | 229 |
| Maximum source length | 128 |
| Learning rate | 1e-3 |
| Dropout | 0.5 |
| Batch size | 300 |
| Total epochs | 10 |

The training data for the relation identifier can be heuristically constructed from WebEdit dataset in a distant supervision manner, which is also the mainstream scheme for dataset construction of relation extraction models. We consider the target texts (revised text) in WebEdit as the inputs, and then use the given facts to match the words in revised texts. We omit the instances where the head word and the tail word are not in the same sentence, because in this situation it is difficult to determine whether these two words have given relations. Furthermore, we construct many negative samples from the positive samples by randomly deleting the words between the head words and the tail words. The relations between the head words and the tail words in negative samples are annotated as "no relation". We add these negative samples into training set and test set to improve the robustness of our relation identifier.

With the above mentioned model and augmented training data, we obtain a robust relation identifier for WebEdit, which can achieve about 93 F1 score in the test set.

### C.3.2 Training Configuration

We reimplement the EncDecEditor [17] and denote it as Seq2Seq. The original paper only report the architecture of FactEditor [17]. Therefore, we have to make our Seq2Seq model have the same size as the FactEditor. The detailed hyperparameter setting is shown as follows:

| | |
|---|---|
| Embedding size | 300 |
| Table Encoder hidden size | 300 |
| Text Encoder hidden size | 300 |
| Decoder hidden size | 600 |
| Maximum source length | 128 |
| Learning rate | 1e-3 |
| Teacher forcing ratio | 0.5 |
| Dropout | 0.5 |
| Batch size | 200 |
| Total epochs | 30 |

Note that we use the same vocabulary as the corresponding relation identifier. Similar to EncDecEditor and FactEditor, our Seq2Seq model also employs attention and copy mechanism.

### C.4 Data-to-Text Generation

In this part, we describe more details about the experiments of data-to-text generation task.

### C.4.1 Entity and Relation Prediction

The settings of data-to-text generation is similar with that of fact-based text editing. The major difference is that the triplets given in WebNLG dataset cannot directly serve as relation constraints. The entities given in the source inputs may not has the same form in the references. Therefore, similar to the dependency prediction in sentence summarization task, we need to predict whether an entity should appear in its output.

In this paper, we directly use the encoder of T5 [34] to predict how to preserve the entities. Specifically, for all the triplets $(h, w, \tau)$, we transform them into a sequential form. Then we perform the sequence labeling task to predict which entities should be preserved in the output. This prediction task can be optimized jointly with the standard seq2seq learning. It must be pointed out that according to our experiment result, this prediction task itself does not influence the performance of data-to-text generation, so adding this task is not the reason for our performance gains. We can obtain the entities and their corresponding relations and set them as relation constraints during the inference stage by adopting RESEAL, which achieves better relation preservation and enhance the performance.

### C.4.2 Relation Identifier for Data-to-Text Generation

The relation identifier on WebNLG dataset shares the same architecture with the one used in fact-based text editing. The only difference is the vocabulary. Due to the fact that we use T5 as our backbone for WebNLG, we adopt the tokenizer used in T5 to make the tokenizer consistent. The vocabulary size is 32,128.

### C.4.3 Training Configuration

For the training settings for the WebNLG dataset, we train the T5-small and T5-base with a learning rate of 5e-5 and a batch size of 20 for 30 epochs. We optimize the model by Adam [19]. No weight decay and label smoothing are applied. The maximum length for both source and target are 384.

# D  Limitation Discussions

## D.1  Impact of the Relation Identifier

For dependency placement task, there are two types of dependency parser. The left-to-right parser [9] is used for decoding. `Stanza` [32] and `spaCy` are used for the evaluation of algorithm.

For the parser which is used for decoding, the result will benefit from a more accurate parser. For the parser which is used for the evaluation, a more accurate one would increase the absolute values of UC and LC. For more precise evaluation, one can use multiple such parsers to produce an average result.

## D.2  Time Complexity Analysis

Similar to most of decoding-only LCD algorithms, RESEAL is usually slower than the standard beam search, but the overall runtime is within an acceptable range. The bottleneck of decoding speed is the times of decoder forward propagation. For a sentence of length $N$, beam search and DBA take $N$ times forward. Our method should be slower due to the relation identifier. There are at most $2|C|$ candidate sentences to be parsed at time step $t$. Therefore, it will at most take $N(1 + 2|C|)$ times of forward for RESEAL, which still has $O(N)$ time complexity. A faster implementation of RESEAL is to save the outputs and hidden states obtained by the relation identifier at time step $t$, and reuse them at next time step $t + 1$. This implementation is left for the future work.

## D.3  External Dependence Issue

Our proposed RESEAL relies on the external relation identifier. If the relation identifier is poor, it would somewhat mislead the generation process and hurt the overall generation quality anyway. However, at least in our experiment, the relation identifier is of high quality, and we believe that in the tasks of this work, it's not so difficult to train a relatively strong relation identifier.

For dependency placement and sentence summarization tasks, we use the left-to-right dependency parser [9] as the relation identifier. Training the parser is easier and faster than training a language model. The training data is from the English Universal Dependencies Treebank, which is publicly available. The parser achieves 90.93 UAS and 88.99 LAS on the test set of this benchmark, which shows its effectiveness for predicting dependency relations.

For fact-based text editing and data-to-text tasks, we use a biaffine attention model [8] as the relation identifier. We use a single layer LSTM with single direction to extract features. We use two separate MLPs with one hidden layer to extract the features of heads and tails, respectively. Finally, we use a biaffine attention module to classify the relation types. Training this model is also easier and faster than training a language model. The training data can be heuristically constructed from the training set of these two tasks: we consider the target sentences as the inputs, and then use the given fact triple (head, relation, tail) to match the words in the target sentences. The relation identifier reaches 93.14 F1 score on the test set, which shows its effectiveness for predicting factual relations.

Thus we believe that all the relation identifiers used in our work are strong enough to produce high-quality results. Training a stronger relation identifier would definitely enhance the performance, but that would be outside the scope of our paper because our paper focuses on the text generation, not the relation extraction task.

# E  Case Study

Table 8 shows more examples for the dependency placement task.

# F  Impact Statement

Our work has introduced a generic decoding method for Relation-Constrained Decoding (RCD). Similar to most of text generation techniques, our proposed RESEAL has a potential risk of being deployed to generate human-like fake text. We suggest that the users or the programmers can utilize the relation constraints to carefully control the generation, e.g., avoiding generating the wrong facts. Therefore, we still believe that the societal impacts of RESEAL are limited and under control.

Table 8: More examples.

| Method | Generated Sentences | Predicted/Gold Dependencies |
|---|---|---|
| Base | If you 've never tried a brick , try the burger . | ['try', 'burger', 'obl'] |
| DBA | I 'm going to try the burger and the brick . | ['try', 'burger', 'obl'] |
| Rerank | If you 're looking for a good burger , try the brick . | ['try', 'burger', 'obl'] |
| RESEAL | I would not try this brick for a burger . | ['try', 'brick', 'obj', ], ['try', 'burger', 'obl'] |
| Ref | Along with the great burger try a brick of onion rings if you are with someone . | ['try', 'brick', 'obj', ], ['try', 'burger', 'obl'] |
| Base | I 'll come back to work when I call . | None |
| DBA | I 'll come back to work when I call . | None |
| Rerank | I 'll come back to work when I call . | None |
| RESEAL | I 'll come back on a call when I 'm back at work . | ['come', 'call', 'obl'], ['back', 'work', 'obl'] |
| Ref | Ps I may have to come back to work for a 615 call | ['come', 'call', 'obl'], ['back', 'work', 'obl'] |
| Base | I understand that you do not have the approval to leave , but I would like to know if you can give me an update on this . | ['understand', 'ccomp', 'have'], ['have', 'obj', 'approval'] |
| DBA | I understand that you do not have the approval of all of the parties , but I would like to leave it at that . | ['understand', 'ccomp', 'have'], ['have', 'obj', 'approval'] |
| Rerank | I understand that you do not have the approval of the President of the United States , but I would like to leave it at that . | ['understand', 'ccomp', 'have'], ['have', 'obj', 'approval'] |
| RESEAL | I understand that you do not have the approval of all of the parties , but I will leave it at that . | ['understand', 'ccomp', 'have'], ['have', 'obj', 'approval'], ['understand', 'conj', 'leave'] |
| Ref | I understand you may not have credit approval yet so perhaps we can leave the appropriate sections blank in the meantime . | ['understand', 'ccomp', 'have'], ['have', 'obj', 'approval'], ['understand', 'conj', 'leave'] |
| Base | At the time , there were no clients in the facility and the staff rushed out of the building as soon as there was a problem . | ['clients', 'facility', 'nmod'] |
| DBA | The clients of this facility were rushed out of the building by the time I got there and there was no one there to help me . | ['clients', 'facility', 'nmod'], ['rushed', 'time', 'obl'], ['rushed', 'was', 'advcl'] |
| Rerank | The clients of this facility were rushed out of the building by the time I got there and there was no one there to help me . | ['clients', 'facility', 'nmod'], ['rushed', 'time', 'obl'], ['rushed', 'was', 'advcl'] |
| RESEAL | When there was clients in facility , they rushed out of the building at the same time as the clients were leaving . | ['was', 'clients', 'nsubj',], ['rushed', 'time', 'obl', ], ['rushed', 'was', 'advcl'], ['clients', 'facility', 'nmod'] |
| Ref | Overpriced and the doctor acted arrogant and rushed at a time when there was very few clients in the facility . | ['was', 'clients', 'nsubj',], ['rushed', 'time', 'obl', ], ['rushed', 'was', 'advcl'], ['clients', 'facility', 'nmod'] |