# OpenReview forum: "Relation-Constrained Decoding for Text Generation"
_NeurIPS.cc/2022/Conference — NeurIPS 2022 Accept_

### Official Review · Reviewer_bPF5 · 2022-07-09

**Rating:** 5
**Confidence:** 5
**Soundness:** 3 good
**Presentation:** 3 good
**Contribution:** 3 good

**Summary:**

The paper is under the theme of constrained decoding, which incorporates predefined rules into a generated sentence. Compared with previous approaches mainly applied to words or phrases, the authors focus on the relations among them, which are called as Relation-Constrained Decoding (RCD). In terms of this setting, they introduce a model, RESEAL, that contains two special modules: RElation-guided probability Surgery and bEam ALlocation, which respectively modify the emission probabilities from the decoder and utilize the constraint information to guide the candidate selection of beam search.

**Questions:**

1, May you put some case studies to show how your model works in practice?
2, Eq. (3) only considers past information; What about the future? Does this scoring style have a bias in this sense?
3, It seems that applying p_{rel) in every decoding step is not reasonable. Could you explain more about this, especially the intuition of g?

**Limitations:**

I've mentioned previously in 'Strengths And Weaknesses': 1) No big difference between the proposed model and DBA except that the former applies external knowledge to change the emission probabilities of the decoder, which is a bit trivial; 2) the reliance of the introduced model on external knowledge.

Will improve my score if the authors can address my concerns.


**Strengths And Weaknesses:**

Pros:
1) well-defined problem that is worth investigation;
2) experiments on many tasks, which seems to be solid.

Cons:
1) the writing of Sec. 3 needs to be polished. Also please don't use python codes in your Algorithm 2;
2) the model seems to be incremental to DBA and is not flexible since it resorts to external knowledge (e.g., a well-trained dependency parser); I also think the authors should even clarify the external dependence issue in the introduction, which is the main problem we are facing in the NLP industry.

---

> ### Author Response · Authors · 2022-08-02
> **Response to Reviewer bPF5 (Part 2/2)**
>
> > 5. "Eq. (3) only considers past information; What about the future? Does this scoring style have a bias in this sense?"
>
> Thanks for raising this interesting concern. In our work, we do focus on the past information, since our method is designed for models with **autoregressive** decoders in the left-to-right manner. **Our method can be extended to consider future information.** We can lookahead to see the decoding result in the following several steps, and calculate the $p_{rel}$ by Eq.(3). However, in this sense, the decoding speed would be slower and the computational cost will be higher. Thus, we mainly focus on the previous generated sequence to do probability surgery to reach a good trade-off between generation efficiency and quality. You have proposed a good point in the question, and we will explore this in the future work.
>
> > 6. "It seems that applying p_{rel} in every decoding step is not reasonable. Could you explain more about this, especially the intuition of g?"
>
> **First of all, $p_{rel}(w|y_{<t},C)$ indicates the probability that the token $w$ satisfies the relation constraints $C$ given previous decoding result $y_{<t}=(y_1,...,y_{t-1})$**. The $p_{rel}$ serves as an external signal, so the model can be aware of whether the relation constraints are satisfied.
>
> **We aim to design an approach to combine $p_{rel}$ and $p_{vocab}$ (the next token probability predicted by the model) to produce an augmented distribution $\widetilde{p}$.** There are many ways to combine these two distributions. In this paper, we use the form $\widetilde{p}\propto g(p_{rel})\cdot p_{vocab}$ (Eq.(4)), where $g$ is a gate function (Eq.(5)) ranging from 0 to 1.
>
> **About the intuition of $g$**, we have introduced in Section 3 (line 132-142) and Appendix A. We summarize them as follows:
> - If adding $w$ at the end of $y_{<t}$ violates the relation constraints, $p_{vocab}$ of $w$ should be reduced. If adding $w$ at the end of $y_{<t}$ satisfies the relation constraints, or w is not included in the constraints, $p_{vocab}$ of $w$ should keep unchanged. (There is an alternative that directly increasing the $p_{vocab}$ if satisfying constraints. However, the experiment shows the alternative will make all the relation constraints be satisfied at the very beginning, thus affecting the overall generation quality.)
> - The function $g$ should not be a monotonically decreasing function, because we want the weight $g(p_{rel})$ to increase when $p_{rel}$ increases.
> - The output of function $g$ should not be zero. Outputting zero will result in negative infinite log-likelihood.
> - A threshold mechanism can be introduced to make the function form more flexible. If the $p_{rel}$ is larger than the threshold $\rho$, it can be considered as satisfying relation constraints.
>
> Therefore, we find the form of $g$ introduced in our paper can meet all the above-mentioned requirements. We adopt this form and find it very effective for our proposed RESEAL.
>
> With the above-mentioned definitions, **we can explain why we can apply $p_{rel}$ in every decoding step to solve the RCD problem**. Specifically:
>
> (1) Applying $p_{rel}$ in every step serves as a stronger signal to guide the relation-constrained generation. This enables to model to dynamically adjust the word probability during the generation process.
>
> (2) Applying $p_{rel}$ in every step can "early stop" candidates which violate relation constraints. For example, if a candidate sentence has unexpected relations or there are no relations between two specific words, $p_{rel}$ will be close to zero. Then this candidate sentence with low $p_{rel}$ will have a lower final score according to our Eq.(4). It would be more likely to be filtered out at early stage of decoding, which is more effective.

---

> ### Author Response · Authors · 2022-08-02
> **Response to Reviewer bPF5 (Part 1/2)**
>
> We sincerely appreciate your constructive comments. We respond to your main concerns below:
>
> > 1. "the writing of Sec. 3 needs to be polished. Also please don't use python codes in your Algorithm 2"
>
> Thanks for pointing out this. We improve our writing of Section 3 and pseudo codes in Algorithm 2. To check more details, please refer to our rebuttal revision version of paper.
>
> > 2. The model seems to be incremental to DBA / No big difference between the proposed model and DBA except that the former applies external knowledge to change the emission probabilities of the decoder, which is a bit trivial
>
> (1) First of all, our work has defined a completely novel decoding scenario, i.e., Relation-Constrained Decoding (RCD). As you have mentioned, RCD is an important problem that is worth investigation. However, the original DBA is designed specifically for lexically constrained decoding, which can not solve the problem of RCD properly. Therefore, it is actually very important to make these seemingly incremental improvements to tackle the problem of RCD. That is, to the best of our knowledge, the first of this kind of contribution.
>
> (2) As Reviewer KEQA have noted, our work has presented a novel technique that can introduce more control of the decoding process by incorporating relations. This can not be achieved by DBA, and is a novel contribution.
>
> (3) Moreover, we thoughtfully design the RESEAL (refer to our response for your Question 6). RESEAL achieves promising improvement on three down stream tasks (e.g., +1.27 ROUGE-L on average for sentence summarization, +4.59 SARI for fact-based text editing, +0.42 BLEU over T5-base for data-to-text generation), without any changes in the training stage. We only modify the decoding stage and achieve improvements on the same base model. That is, an important contribution of this work.
>
> That we could make those non-trivial efforts seem effortless, we’ll take this as a compliment:)
>
> > 3. "The reliance of the introduced model on external knowledge." / "I also think the authors should even clarify the external dependence issue in the introduction, which is the main problem we are facing in the NLP industry."
>
> This external dependence issue is also the concern of Reviewer brwW and Reviewer KEQA, so we add an official comment titled **"Response to External Dependence Issue"** on top of this webpage to explain this issue in detail. Please refer to that for more information, thanks!
>
> We have modified our introduction part to include such content. Thank you for your advice!
>
>
> > 4. "May you put some case studies to show how your model works in practice?"
>
> Sure. We believe that our model has many (potential) applications in practical scenarios, e.g., controllable text generation, improving the factual consistency in summarization or dialogue generation, or some data-to-text applications.
>
> We also have provided some examples in Appendix D.3. We will move them to the main part of paper in the revised version with more pages available. We list some examples for your reference:
>
> - **[Relation Constraints]** (thought, 'ccomp', ’s), (’s, 'nsubj', charges)
>
> **[DBA]** I **thought** the **charges** would be $ 10,000, but it **’s** $ 20,000.
>
> **[RESEAL]** I **thought** there **’s** some **charges**, but I was wrong.
>
> **[Reference]** I always **thought** there **’s** no custom **charges** for gifts.
>
> - **[Relation Constraints]** (was, 'nsubj', clients), (rushed, 'obl', time), (rushed, ‘advcl’, was), (clients, ‘nmod’, facility)
>
> **[DBA]** The **clients** of this **facility** were **rushed** out of the building by the **time** I got there and there **was** no one there to help me .
>
> **[RESEAL]** When there **was clients** in **facility** , they **rushed** out of the building at the same **time** as the clients were leaving .
>
> **[Reference]** Overpriced and the doctor acted arrogant and **rushed** at a **time** when there **was** very few **clients** in the **facility** .
>
> In the first example, DBA misses both the (thought, 'ccomp', ’s) and (’s, 'nsubj', charges) constraints. In the second example, DBA misses the (was, 'nsubj', clients) constraints. DBA can satisfy all the lexical constraints, but it often fails to handle the relations between them. RESEAL can produce fluent sentences that are close to the grammatical structure of the references.

---

> ### Author Response · Authors · 2022-08-07
> **Kind reminder to the reviewer**
>
> Dear reviewer bPF5:
>
> We sincerely thank you for the review and comments. We have provided corresponding responses and results, which we believe have covered your concerns. We hope to further discuss with you whether or not your concerns have been addressed. Please let us know if you still have any unclear parts of our work.
>
> Best regards, The Authors

---

> ### Author Response · Authors · 2022-08-09
> **The end of the discussion is approaching**
>
> Dear reviewer bPF5:
>
> We sincerely appreciate your comprehensive and constructive comments. Since the end of the discussion is approaching, it would be great to let us know whether our responses can address your concerns. Could you please also let us know if there are any other concerns that we should address? We would be pleased to clarify them and revise our paper by the response deadline.
>
> Best regards, The Authors

---

### Official Review · Reviewer_KEQA · 2022-07-11

**Rating:** 4
**Confidence:** 3
**Soundness:** 2 fair
**Presentation:** 1 poor
**Contribution:** 2 fair

**Summary:**

This paper presents a way to incorporate "relation constraints" in decoding. For example, given the constraint triplet: (phone, amod, fancy), the decoding process will attempt to generate a sentence with that includes words "phone" and "fancy" while also satisfying an "amod" relation based on a relation predictor (e.g. "This is a fancy phone"). This is an extension of the lexically-constrained decoding problem, though it seems that the relation constraints here are only satisfied probabilistically.

**Questions:**

- In Table 1, it seems that some of the lexically-constrained decoding methods (e.g. DBA) are worse than Base. Is that expected?
- Are there speed comparisons between proposed method and the simple Rerank baseline?
- Can you provide some example sentences of your decoder vs DBA in the main part of the paper? This may help motivate the use of relations.
- It seems to me that the relations in the decoder are always predicted. It is a hidden variable, so it is somewhat different in nature to the words in the lexical constraints, which are observed. If you have a poor relation predictor, can it fool your decoder into believing that the constraints are satifised?

**Limitations:**

Appendix E writes: "... our proposed methods may be used to generate more human-like fake text. However, the impacts are more apparent when considering deployed applications, while our proposed methods as the methodologies can not have any direct negative societal impacts."

I am actually not sure what you are saying here. Do you mean that as a general technique, this does not have direct negative social impact, because the fault is with the application that uses this technique? I guess I see your point partly but I think the paragraph can be modified to say things in a better way. If your method is so good it makes it more difficult to detect fake text, isn't that a direct negative impact compared to other methods?

**Strengths And Weaknesses:**


Strength
- This is an interesting problem. Techniques that give us more control of the decoding process is generally a nice contribution to the field. I am not aware of previous work that incorporates relations.

Weakness:
- While relation contraints are interesting, it is not immediately clear in what applications they are helpful. Various tasks and results are shown, but I don't see intuitively why the relation constraint is important. For example, given the constraint triplet: (phone, amod, fancy), would a simple lexically-constraint decoder that uses (phone, fancy) produce those words in the right order anyway (due to a strong language model)? In contrast, would your relation-constrained decoder produce these words in the wrong order but still labeled with the right head-child dependency relation (due to a poor relation predictor)? Similarly, for the summarization problem, why do relations help? Perhaps the Data-to-Text and Fact-based Text editing tasks are easier to motivate, but the paper doesn't dedicate enough prose explaining these tasks.

- The proposed method is difficult to understand. I think I got it after several readings, but there may be ways to improve especially the understanding of the general purpose of line 4 and 5 in Algo 1. One suggestion is to elaborate on the left part of Figure 1 (which explains line 4 at least).

---

> ### Author Response · Authors · 2022-08-02
> **Response to Reviewer KEQA (Part 3/3)**
>
> > 7. "Can you provide some example sentences of your decoder vs DBA in the main part of the paper? "
>
> Sure. We have provided some examples in Appendix D.3. We will move them to the main part of paper in the revised version with more pages available.
>
> We list some examples for your reference:
>
> (1) **[Relation Constraints]** (thought, 'ccomp', ’s), (’s, 'nsubj', charges)
>
> **[DBA]** I **thought** the **charges** would be $ 10,000, but it **’s** $ 20,000.
>
> **[RESEAL]** I **thought** there **’s** some **charges**, but I was wrong.
>
> **[Reference]** I always **thought** there **’s** no custom **charges** for gifts.
>
> (2) **[Relation Constraints]** (was, 'nsubj', clients), (rushed, 'obl', time), (rushed, ‘advcl’, was), (clients, ‘nmod’, facility)
>
> **[DBA]** The **clients** of this **facility** were **rushed** out of the building by the **time** I got there and there **was** no one there to help me .
>
> **[RESEAL]** When there **was clients** in **facility** , they **rushed** out of the building at the same **time** as the clients were leaving .
>
> **[Reference]** Overpriced and the doctor acted arrogant and **rushed** at a **time** when there **was** very few **clients** in the **facility** .
>
> In the first example, DBA misses both the (thought, 'ccomp', ’s) and (’s, 'nsubj', charges) constraints. In the second example, DBA misses the (was, 'nsubj', clients) constraints. DBA can satisfy all the lexical constraints, but it often fails to handle the relations between them. RESEAL can produce fluent sentences that are close to the grammatical structure of the references.
>
> > 8. "It seems to me that the relations in the decoder are always predicted. It is a hidden variable, so it is somewhat different in nature to the words in the lexical constraints, which are observed. If you have a poor relation predictor, can it fool your decoder into believing that the constraints are satifised?"
>
> We agree that many existing works view the relations as a hidden variable. This is an **implicit** way of predicting relations. However, that is exactly the main difference between our work and these existing works. We propose an **explicit** way of handling the relations, which can somewhat improve the **interpretability and controllability** of the models.
>
> About the "poor relation predictor", we have addressed this issue in the official comment on the top.
>
> > 9. About social impact of our method
>
> We're sorry for this unclear section. We have modified the Appendix E as follows:
>
> "Our work has introduced a generic decoding method for Relation-Constrained Decoding (RCD). Similar to most of text generation techniques, our proposed RESEAL has a potential risk of being deployed to generate human-like fake text. We suggest that the users or the programmers can utilize the relation constraints to carefully control the generation, e.g., avoiding generating the wrong facts. Therefore, we still believe that the societal impacts of RESEAL are limited and under control. "

---

> ### Author Response · Authors · 2022-08-02
> **Response to Reviewer KEQA (Part 2/3)**
>
> > 4. "The proposed method is difficult to understand. I think I got it after several readings, but there may be ways to improve especially the understanding of the general purpose of line 4 and 5 in Algo 1. One suggestion is to elaborate on the left part of Figure 1 (which explains line 4 at least)."
>
> Thanks for your suggestion. Actually, in the left part of Figure 1, we have shown the detailed processes of line 4 in Algo 1. In Figure 1, we showcase how our RESEAL works with an example of RCD. In this case, the given relation constraints are (phone, amod, fancy). At decoding time step 4, let us consider two candidates in the beam search, i.e., "What a fancy" (denoted as $Y_1$) and "This phone is" (denoted as $Y_2$). In standard decoding manner, model will output a next-token probability $p_{\text{vocab}}$ according to the context. The next-token probability for $Y_1$ is shown in the top of Figure 1 (denoted as $p_{\text{vocab}}^{(1)}$), and the next-token probability for $Y_2$ is shown in the bottom of Figure 1 (denoted as $p_{\text{vocab}}^{(2)}$). Note that $p_{\text{vocab}}^{(2)}$ includes the probability of the gray part.
>
> In our work, RESEAL will operate the produced probability distributions according to the result $p_{rel}$ of a relation identifier. Since the token "phone" can form a "amod" relation with the token "fancy" in $Y_1$, which satisfies the given relation constraints, the relation identifier will predict a high $p_{rel}$ for "phone" in $p_{\text{vocab}}^{(1)}$. On the contrary, the token "fancy" will form a wrong relation "nsubj" with the token "phone" in $Y_2$. Therefore, the $p_{rel}$ for "fancy" in $p_{\text{vocab}}^{(2)}$ will be relatively low. Then, according to Equation 5, when calculating the augmented distribution $\tilde{p}$, RESEAL will preserve the probability of "phone" in $p_{\text{vocab}}^{(1)}$, while it will reduce the probability of "fancy" in $p_{\text{vocab}}^{(2)}$ (as shown in Figure 1, the probability of "fancy" in $p_{\text{vocab}}^{(2)}$ is cut down.) This is what line 4 in Algo 1 does.
>
> We will add these explanations in the revised version.
>
>
> > 5. "In Table 1, it seems that some of the lexically-constrained decoding methods (e.g. DBA) are worse than Base. Is that expected?"
>
> We believe that is expected. We list some reasons for your reference:
> - The "Base" is a very strong baseline, because it exploits the strong pre-train language model BART.
> - Some lexical-constrained decoding methods (DBA, DDBA) would **forcibly add the unsatisfied lexical constraints into the candidate word set.** This is the key to ensuring the presence of lexical constraints. However, doing this fails to consider the relations between words, thus would make the generated sentence less fluent.
>
>
> > 6. "Are there speed comparisons between proposed method and the simple Rerank baseline?"
>
> Yes. We discuss the speed of our proposed method in Appendix D.2.
>
> Like most lexically-constrained decoding methods (DBA, NEUROLOGIC), RESEAL is usually slower than the standard beam search and the Rerank baseline, but the overall runtime is within an acceptable range. We summarize the time complexity as follows:
>
> - For a sentence of length $N$, standard beam search takes $N$ times of forward propagation.
> - For Rerank baseline, it takes extra $k$ times of forward to parse the $k$ candidates. Overall, it takes $N+k$ times forward.
> - For DBA, it takes $N$ times of forward. However, DBA needs some extra time to handle the candidate set.
> - For RESEAL, let $|C|$ denote the number of the relation constraints. Thus there are at most $2|C|$ candidate sentences to be parsed at time step $t$. It takes at most take $N(1 + 2|C|)$ times of forward for RESEAL, which still is $O(N)$ time complexity. Moreover, there are many implementation tricks to be applied to optimize the actual decoding speed.

---

> ### Author Response · Authors · 2022-08-02
> **Response to Reviewer KEQA (Part 1/3)**
>
> We sincerely appreciate your constructive comments. We respond to your main concerns below:
>
> > 1. "For example, given the constraint triplet: (phone, amod, fancy), would a simple lexically-constraint decoder that uses (phone, fancy) produce those words in the right order anyway (due to a strong language model)? "
>
> **It's relatively more difficult for lexically-constrained decoder to control the word order.** For the relation constraints (phone, amod, fancy), the lexically-constrained decoder may produce the right order "fancy phone", which may be simply due to **the similar training samples** used in pre-training or fine-tuning stage. It's hard to control this order because the training data distribution is complex.
>
> **With our proposed relation constraints, we can elegantly control the word order in the decoding stage.** With relation constraints (phone, amod, fancy), we can obtain a sentence like "The fancy phone...". With relation constraints (fancy, nsubj, phone), we can obtain a sentence like "This phone is fancy...".
>
> > 2. "Would your relation-constrained decoder produce these words in the wrong order but still labeled with the right head-child dependency relation (due to a poor relation predictor)"
>
> This external dependence issue is also the concern of Reviewer brwW and Reviewer bPF5, so we add an official comment titled **"Response to External Dependence Issue"** on top of this webpage to explain this issue in detail. Please refer to that for more information, thanks!
>
> > 3. "For the summarization problem, why do relations help? Perhaps the Data-to-Text and Fact-based Text editing tasks are easier to motivate, but the paper doesn't dedicate enough prose explaining these tasks."
>
> **For the sentence summarization, relations may help because some dependency relations in the source may be important. These important relations should be preserved in the summary.**
>
> There are some examples from GigaWord dataset (# denotes a specific number after preprocessing):
>
> (1) **[Source]** Vietnam forecasts to gain export revenues of #.# billion US dollars **in** the two remaining **months** of this year , a year-on-year increase of ##.# percent .
>
> **[Summary]** Vietnam to post higher export earnings **in** next # **months**
>
> **[Relation Constraints]** (months, 'case', in)
>
> (2) **[Source]** As the indonesian people 's consultative assembly finished drafting decrees relating to president abdurrahman wahid 's accountability , some **large** political **parties** in the country would hold meeting to narrow possible differences **on** the **agenda** of the special mpr session scheduled for august # .
>
> **[Summary]** Indonesian **large parties** to meet **on** assembly session **agenda**
>
> **[Relation Constraints]** (parties, 'amod', large), (agenda, 'case', on)
>
> In these cases, the key information of the source sentence can be expressed by several relation constraints. If we can preserve these relations, we can produce an informative summary. Our proposed RESEAL enables the model to focus on the important dependency relations explicitly, which helps to improve the summarization quality.
>
> Apart from that, the Data-to-Text and Fact-based Text editing tasks use the relations more directly. We will add some contents explaining these downstream tasks in our revised version.

---

> ### Author Response · Authors · 2022-08-07
> **Kind reminder to the reviewer**
>
> Dear reviewer KEQA:
>
> We sincerely thank you for the review and comments. We have provided corresponding responses and results, which we believe have covered your concerns. We hope to further discuss with you whether or not your concerns have been addressed. Please let us know if you still have any unclear parts of our work.
>
> Best regards, The Authors

---

### Official Review · Reviewer_brwW · 2022-07-12

**Rating:** 7
**Confidence:** 4
**Soundness:** 4 excellent
**Presentation:** 3 good
**Contribution:** 3 good

**Summary:**

This paper proposes a new paradigm in constrained text generation called relation-constrained decoding (RCD). RCD ensures that the tokens in the constraint set are adhered to (lexically constrained decoding) while decoding from NLG models without compromising on the underlying "relation" between the tokens. To address RCD, the paper presents a decoding time algorithm called RESEAL that can handle different types of relations (syntactic, factual, or entity based). On top of the standard evaluation, the paper also highlights results on 3 NLG tasks that help understand the RCD paradigm's applicability.

Implementation of RESEAL involves two main strategies, probability surgery, and relation-guided lexical decoding.

**Questions:**

1. Line 102: Based on the formulation the candidates are just getting re-ranked. How is the signal guaranteeing the presence of relation constraint?
2. Line 130: Why is 2 needed in the denominator?
3. Table 2: Shouldn't w/o prob = DBA? Why is the result different from DBA (Table 1)?
4. Line 225: What is the word+rel strategy?

Typo:
Line 111: Line 1-7 -> Line 1-6

**Limitations:**

The paper highlights some of the limitations in section 5.1(results) and societal impacts in the appendix.

Limitation: The model needs performance depends on access to a good relation identifier. The errors in the relation-identifier propagate into the model.

**Strengths And Weaknesses:**

Strengths:
1. The paper is well written and easy to understand.
2. Comprehensive evaluation of a variety of NLG tasks.
3. Good results against competitive baselines.
4. Fact-based editing is an elegant application of this paradigm.

Weaknesses:
1. Dependence upon a suitable relation identifier is a prerequisite. In most cases, it won't be easy to obtain/train. Thereby limiting the applicability of RCD to very simplistic tasks. In most cases, a lexically constrained decoding paradigm is sufficient.
2. Evaluation on newer and robust NLG evaluators like BERTScore, and BLEURT is missing.
3. No human evaluation of results.

---

> ### Author Response · Authors · 2022-08-02
> **Response to Reviewer brwW**
>
> We sincerely appreciate your constructive comments. We respond to your main concerns below:
>
> > 1. "Dependence upon a suitable relation identifier is a prerequisite. In most cases, it won't be easy to obtain/train. Thereby limiting the applicability of RCD to very simplistic tasks. In most cases, a lexically constrained decoding paradigm is sufficient."
>
> This external dependence issue is also the concern of Reviewer KEQA and Reviewer bPF5, so we add an official comment titled **"Response to External Dependence Issue"** on top of this webpage to explain this issue in detail. Please refer to that for more information, thanks!
>
> About the applicability of RCD, thank you again for pointing out this. We believe that adding more complex constraints (not limited to lexical constraints) is helpful to better control the text generation, and this would be a promising direction. We will explore more potential applications of RCD in future work.
>
> > 2. "Evaluation on newer and robust NLG evaluators like BERTScore, and BLEURT is missing. No human evaluation of results."
>
> We agree that these new metrics and human evaluation are helpful. We will add them in our revised version.
>
> > 3. Based on the formulation the candidates are just getting re-ranked. How is the signal guaranteeing the presence of relation constraint?
>
> The signal is through beam allocation (please refer to line 10-12 of Algorithm 2). We propose to use **the number of correct relation constraints** of i-th candidates (denoted as $n_i$) to divide the banks. The banks determine the way of beam expansion. The larger $n_i$ means less beam expansion.
>
> The overall effect of this operation is that (which is similar to the original DBA):
> - The candidate sentence with a small $n_i$ will get more chances to continue decoding. Therefore it will have more chances to satisfy the rest of the relation constraints.
> - The candidate sentence with a large $n_i$ will get fewer chances to continue decoding, since there are fewer relation constraints to be satisfied, it does not need so many beams to continue decoding.
>
> > 4. Why is 2 needed in the denominator?
>
> It's needed because we have 2 kinds of probabilities $p_{trans}$ and $p_{type}$ in Eq (3).
>
> > 5. Shouldn't w/o prob = DBA? Why is the result different from DBA (Table 1)?
>
> In Table 2, "w/o prob" indicates without probability surgery but still having RG-Top-K. Sorry for the potential misleading. We have fixed this in our revised version.
>
> > 6. What is the word+rel strategy?
>
> The "word+rel" means that we change the meaning of $n_i$ mentioned above. For "word+rel", $n_i$ denotes the number of satisfied lexical constraints + the number of (dependency) relation constraints. DBA only includes the former, while our RESEAL only includes the latter.

---

### Official Review · Reviewer_hFXg · 2022-07-13

**Rating:** 6
**Confidence:** 4
**Soundness:** 3 good
**Presentation:** 3 good
**Contribution:** 3 good

**Summary:**

The paper describes a model for text generation, based on target dependency relations that should be in the output.
The word-level output probabilties are modified to increase the likelihood of generating words that match the target relation.
During beam decoding, the candidate construction method also takes the target relations into account.
Evaluation is performed on several datasets, formulating the task as text generation based on dependency relations.

**Questions:**

It is currently unclear how many dependencies are given as input to the model. And if they are not all the dependencies, then how are they chosen.

Table 8 in the appendix seems to have some issues. The dependencies in the first row do not match the sentences.


**Limitations:**

There is a very brief discussion of computational complexity in the appendix. There could be a much more thorough discussion on the limitations of the system, the task, the evaluation metrics and text generation based on dependencies in general.

**Strengths And Weaknesses:**

The model is interesting.
The paper is well written and clear.
The specific method of modifying the output probabilities based on the given relations can be useful.
Results are demonstrating the benefit of the proposed model over the chosen baselines.

The work is presented as a completely novel task. However, it does not sufficiently address existing work on data-to-text generation.
In data-to-text, the task is very similar - generation text based on relation tuples.
While data-to-text systems do not typically restrict themselves to only dependency relations, they could definitely be applied on this task.
Therefore, work in this area should be covered under related research and relevant baselines should be used for comparison.
There is currently a very minimal section for evaluation on a data-to-text dataset, but only a very generic model (not designed for data-to-text) is used as a baseline.

There is a nice section on dependency-guided generation, which has the same goal as the proposed task. However, none of the listed previous work seem to be reported as a baseline.

The proposed evaluation metrics are interesting for diagnostics but overall not very convincing.
It is claimed that overlap-based metrics are not appropriate and parser-based metrics are used instead.
This doesn't really reflect how readable the sentence is at all.
For a high score, the model could just linearise the triplets that are provided as input, without producing a coherent sentence.
The UC and LC metrics only measure recall, not precision, so there is no penalty for generating loads of different relations in the output.
Similarly, the Word% metric could just copy over the model input into the output in order to get a 100% score.
BLEU 1-4 should be reported, along with METEOR. One BLEU is currently reported but it is unclear which one it is. BLEU-1 would not be sufficient to measure sentence quality.

---

> ### Author Response · Authors · 2022-08-02
> **Response to Reviewer hFXg**
>
> We sincerely appreciate your constructive comments. We address your concerns below:
>
> > 1. "It does not sufficiently address existing work on data-to-text generation ... Therefore, work in this area should be covered under related research and relevant baselines ... is used as a baseline."
>
> We are sorry that we have included the evaluation on the data-to-text task in such a minimal section due to the page limit. Thanks for your suggestion, and we have additionally included the task-specific baselines in the following table:
>
> | Models       | BLEU-4 |
> | ------------ | --------- |
> | [1] Castro Ferreira et al. (2019)   | 51.68  |
> | [2] Moryossef et al. (2019)  | 47.24   |
> | [3] Zhao et al. (2020a)    | 52.78     |
> | [4] Harkous et al. (2020) | 52.90    |
> | [5] Nan et al. (2021) | 45.89   |
> | T5-small | 56.34 | 42.78 |
> | T5-small+RESEAL  | 56.87  |
> | T5-base     | 59.17    |
> | T5-base+RESEAL  | **59.59**  |
>
> It is worth to mention that our proposed RESEAL can be adapted to any model as a plug-and-play decoding algorithm. In our experiments, we adapt RESEAL to the current SOTA model i.e. T5, and observe a further improvement. This speaks volumes for the effectiveness of our approach. We will add these results in the revised version with more pages available.
>
> > 2. There is a nice section on dependency-guided generation, which has the same goal as the proposed task. However, none of the listed previous work seem to be reported as a baseline.
>
> As you have mentioned, our dependency placement is a completely novel task, so actually there are no existing baselines directly designed for this task. We have listed some works related to dependency-guided generation to illustrate the effect of dependency for text generation. Some of them are designed for specific tasks, and others cannot directly used for dependency placement task. Moreover, it's worth noting that SemSum[6], which has been mentioned in "dependency-guided generation" section, is reported as one of the baseline for sentence summarization.
>
> > 3. BLEU 1-4 should be reported, along with METEOR. One BLEU is currently reported but it is unclear which one it is. BLEU-1 would not be sufficient to measure sentence quality.
>
> **All the BLEUs in our paper are BLEU-4**. We will clarify this in the revised version.
>
> Then we agree that more evalution metrics should be reported. Thanks for your suggestions. We will report them in the following table:
>
> | Models       | BLEU-1 | BLEU-2 | BLEU-3 | BLEU-4 | METEOR |
> | ------------ | --------- | --------- | --------- | --------- | --------- |
> | Base   | 40.09 | 25.78 | 17.40 | 11.92 | 20.12 |
> | Rerank ($k=20$) | 39.58 | 25.38 | 17.09 | 11.66 | 20.18 |
> | CGMH | 23.04 | 8.30 | 3.30 | 1.47 | 14.50 |
> | X-MCMC | 30.62 | 15.03 | 8.20 | 4.62 | 17.04 |
> | X-MCMC-C | 32.28 | 17.50 | 10.48 | 6.39 | 17.65 |
> | DBA | 39.93 | 25.33 | 16.93 | 11.47 | 20.12 |
> | DDBA | 41.08 | 26.34 | 17.75 | 12.22 | 20.12 |
> | NEUROLOGIC | 41.52 | 26.75 | 17.98 | 12.23 | 20.13 |
> | RESEAL (this work) | **43.79** | **27.81** | **18.64** | **12.62** | **20.40** |
>
> > 4. It is currently unclear how many dependencies are given as input to the model. And if they are not all the dependencies, then how are they chosen.
>
> Our Table 6 (please refer to the Appendix B) reports how many dependencies are given as input to the model:
>
> |       | train | dev | test |
> | ------------ | --------- |  --------- | --------- |
> | Avg number of dependencies | 2.43 | 2.04 | 2.04 |
> | Max number of dependencies | 18 | 13 | 11 |
>
> They are not all the dependencies of the sentences. We select them by random sampling (at a ratio of 40%). Moreover, we only consider the dependency relations whose modifier is an adjective, noun, or verb.
>
> For more details about the dataset construction process, please refer to Appendix B. Thanks!
>
> > 5. Table 8 in the appendix seems to have some issues.
>
> We're sorry for this mistake, and we have fixed them in our revised version.
>
> > 6. A much more thorough discussion on the limitations
>
> Thank you for your suggestions. We have added a new section in the Appendix discussing the limitations of our work.
>
>
> [1] Castro Ferreira et al. "Neural data-to-text generation: A comparison between pipeline and end-to-end architectures". EMNLP/IJCNLP (1) 2019: 552-562
>
> [2] Moryossef et al. "Step-by-Step: Separating Planning from Realization in Neural Data-to-Text Generation". NAACL-HLT (1) 2019: 2267-2277
>
> [3] Zhao et al. "Bridging the Structural Gap Between Encoding and Decoding for Data-To-Text Generation". ACL 2020: 2481-2491
>
> [4] Harkous et al. "Have Your Text and Use It Too! End-to-End Neural Data-to-Text Generation with Semantic Fidelity". COLING 2020: 2410-2424
>
> [5] Nan et al. "DART: Open-Domain Structured Data Record to Text Generation". NAACL-HLT 2021: 432-447
>
> [6] Jin et al. "Semsum: Semantic dependency guided neural abstractive summarization". AAAI 2020.

---

### Author Response · Authors · 2022-08-02
**Response to External Dependence Issue**

We thank the reviewers for their constructive comments. The external dependence issue is the common concern of the reviewers, therefore it should be first answered in this section.

We agree with the reviewers that our RESEAL relies on the external relation identifier (or say "relation predictor"). If the relation identifier is poor, it would somewhat mislead the generation process and hurt the overall generation quality anyway.

**However, at least in our experiment, the relation identifier is of high quality, and we believe that in the tasks of this work, it's not so difficult to train a relatively "strong relation predictor".** Specifically:
- For dependency placement and sentence summarization tasks, we use the left-to-right dependency parser[1] as the relation identifier. Training the parser is easier and faster than training a language model. The training data is from the English Universal Dependencies Treebank, which is publicly available. **The parser achieves 90.93 UAS and 88.99 LAS on the test set of this benchmark, which shows its effectiveness for predicting dependency relations.**
- For fact-based text editing and data-to-text tasks, we use a biaffine attention model[2] as the relation identifier. We use a single layer LSTM with single direction to extract features. We use two separate MLPs with one hidden layer to extract the features of heads and tails, respectively. Finally, we use a biaffine attention module to classify the relation types. Training this model is also easier and faster than training a language model. The training data can be heuristically constructed from the training set of these two tasks: we consider the target sentences as the inputs, and then use the given fact triple (head, relation, tail) to match the words in the target sentences. **The relation identifier reaches 93.14 F1 score on the test set, which shows its effectiveness for predicting factual relations.**

Thus we believe that all the relation identifiers used in our work are strong enough to produce high-quality results. **Training a stronger relation identifier would definitely enhance the performance, but that would be outside the scope of our paper because our paper focuses on the text generation, not the relation extraction task.**

[1] Fernández-González, Daniel, and Carlos Gómez-Rodríguez. "Left-to-Right Dependency Parsing with Pointer Networks." NAACL-HLT (1). 2019.

[2] Dozat, Timothy, and Christopher D. Manning. "Deep Biaffine Attention for Neural Dependency Parsing." ICLR. 2016.

---

### Author Response · Authors · 2022-08-08
**The end of the discussion phase approaching.**

Dear Reviewers,

There are less than 48 hours until the end of the discussion phase (09 Aug). Could you please go over our responses and the revision so we can have more discussions? We have responded to your comments and faithfully reflected them in the revised version. We are wondering whether your concerns have been properly addressed.

We sincerely thank you for your time and efforts in reviewing our paper, and for your insightful and constructive comments.

Best regards, The Authors

---

### Author Response · Authors · 2022-08-09
**Summary of the Revision**

We really appreciate all the reviewers for their efforts in reviewing our paper. We believe that we have responded to most of the concerns below. Since the discussion deadline is approaching, here we'd like to briefly summarize the updates we have made to the revised version of the paper:

- We have clarified the external dependence issue in the introduction and added an appendix (Sec. E) to further discuss it, as suggested by the reviewers.
- We have explained why RCD can help the downstream tasks in line 242-244, as suggested by Reviewer KEQA.
- We have modified the social impact part, as suggested by Reviewer KEQA.
- We have fixed some typos, as suggested by Reviewer brwW.
- We have polished Sec. 3 (Methodology) and remove the python codes in Algorithm 2, as suggested by Reviewer bPF5.
- We have noted that "Every BLEU reported in this paper is BLEU-4" in the paper, as suggested by Reviewer hFXg.
- We have modified the caption of Table 2 to make it more clear, as suggested by Reviewer brwW.
- Other experimental results (BLEU 1-4, BERTScore, BLEURT, human evaluation, data-to-text baselines) will be added with more pages available.

All the modified parts are marked as red color.

---

### Meta-Review · Area_Chair_5SyG · 2022-08-29

**Recommendation:** Accept
**Confidence:** Less certain

**Metareview:**

The paper describes a model for text generation, based on target dependency relations that should be in the output. The word-level output probabilities are modified to increase the likelihood of generating words that match the target relation. Evaluation is performed on several datasets, formulating the task as text generation based on dependency relations.
The empirical gains are OK but not particularly large. What I find more compelling is the ability to control the output of the model, which is currently lacking in most approaches.
The reviewer scores straddle the decision boundary and it was unfortunately not possible to get the reviewers to engage in a discussion but the authors did a good job addressing all initial comments/questions.

**Award:**

No

---

### Decision · Program_Chairs · 2022-09-14

Accept